# Evaluation of the Anti-*Toxoplasma gondii* Efficacy, Cytotoxicity, and GC/MS Profile of *Pleopeltis crassinervata* Active Subfractions

**DOI:** 10.3390/antibiotics12050889

**Published:** 2023-05-10

**Authors:** Jhony Anacleto-Santos, Fernando Calzada, Perla Yolanda López-Camacho, Teresa de Jesús López-Pérez, Elba Carrasco-Ramírez, Brenda Casarrubias-Tabarez, Teresa I. Fortoul, Marcela Rojas-Lemus, Nelly López-Valdés, Norma Rivera-Fernández

**Affiliations:** 1Departamento de Microbiología y Parasitología, Facultad de Medicina, Universidad Nacional Autónoma de México (UNAM), Ciudad Universitaria, Mexico City 04510, Mexico; anacletosantosjhony@facmed.unam.mx (J.A.-S.); tere.lopez82@comunidad.unam.mx (T.d.J.L.-P.); elba.carrasco@facmed.unam.mx (E.C.-R.); 2Unidad de Investigación Médica en Farmacología, Unidad Médica de Alta Especialidad, Hospital de Especialidades Centro Médico Nacional Siglo XXI, Instituto Mexicano del Seguro Social, Col. Doctores, Cuauhtémoc 06725, Mexico; fernando.calzada@imss.gob.mx; 3Unidad Cuajimalpa, Departamento de Ciencias Naturales, Universidad Autónoma Metropolitana (UAM), Cuajimalpa 05348, Mexico; plopezc@cua.uam.mx; 4Departamento de Biología Celular y Tisular, Facultad de Medicina, Universidad Nacional Autónoma de México (UNAM), Ciudad Universitaria, Mexico City 04510, Mexico; bcasarrubiast@gmail.com (B.C.-T.); fortoul@unam.mx (T.I.F.); marcelarojas@comunidad.unam.mx (M.R.-L.); nellylopez@facmed.unam.mx (N.L.-V.)

**Keywords:** *Toxoplasma gondii*, *Pleopeltis crassinervata*, anti-*Toxoplasma*, terpenoids, fatty acids, gas chromatography

## Abstract

*Pleopeltis crassinervata* (Pc) is a fern that, according to ethnobotanical records, is used in Mexican traditional medicine to treat gastrointestinal ailments. Recent reports indicate that the hexane fraction (Hf) obtained from Pc methanolic frond extract affects *Toxoplasma gondii* tachyzoite viability in vitro; therefore, in the present study, the activity of different Pc hexane subfractions (Hsf) obtained by chromatographic methods was evaluated in the same biological model. Gas chromatography/mass spectrometry (GC/MS) analysis was carried out for hexane subfraction number one (Hsf1), as it showed the highest anti-*Toxoplasma* activity with a half-maximal inhibitory concentration (IC_50_) of 23.6 µg/mL, a 50% cytotoxic concentration (CC_50_) of 398.7 µg/mL in Vero cells, and a selective index (SI) of 16.89. Eighteen compounds were identified by Hsf1 GC/MS analysis, with the majority being fatty acids and terpenes. Hexadecanoic acid, methyl ester was the most commonly found compound (18.05%) followed by olean-13(18)-ene, 2,2,4a,8a,9,12b,14a-octamethyl-1,2,3,4,4a,5,6,6a,6b,7,8,8a,9,12,12a,12b,13,14,14a,14b-eicosahydropicene, and 8-octadecenoid acid, methyl ester, which were detected at 16.19%, 12.53%, and 12.99%, respectively. Based on the mechanisms of action reported for these molecules, Hsf1 could exert its anti-*Toxoplasma* activity mainly on *T. gondii* lipidomes and membranes.

## 1. Introduction

*Toxoplasma gondii* is considered one of the most important foodborne and waterborne parasites of medical and veterinary importance worldwide; 30% of the world’s population has been infected with *T. gondii*. High seroprevalence is reported especially in Europe, South America, and Africa [1].

Humans can be infected with the parasite by consuming tissue cysts which are located in the brain and skeletal muscles of different animals used for human consumption, mainly sheep and pigs, or fruits, vegetables, or water contaminated with sporulated oocysts (that are liberated with felid feces) [2].

Acute *T. gondii* infection is potentially harmful in pregnant women, immunocompromised patients, and fetuses, as it can cause abortions, encephalitis, and congenital malformations, respectively [3]. Chronic toxoplasmosis is related to behavioral disturbances, neuropsychiatric disorders, and other neurological illnesses, such as Alzheimer’s and Parkinson’s [4,5]. Ocular toxoplasmosis may occur following congenital or acquired transmission and can cause necrotizing retinitis, glaucoma, retinal perivasculitis, and blindness [6].

Toxoplasmosis control continues to rely on pharmacological interventions, as there are currently no vaccines approved for human use, and few advances have been made in the treatment of the disease; therefore, the availability of anti-*Toxoplasma* drugs is limited. Pyrimethamine is the standard treatment for acute toxoplasmosis; however, it is not effective when administered alone and must be used in combination with sulfadiazine. This combination causes bone marrow suppression and does not completely eliminate the infection; consequently, treatment is given for long periods. Anti-*Toxoplasma* drugs are effective only against tachyzoite stages (acute toxoplasmosis); hence, tissue cysts remain latent and can reactivate into tachyzoites in immunocompromised patients, and drug resistance has been recently described in atypical *T. gondii* strains [7,8]. Based on these facts, other options to treat toxoplasmosis are desirable.

Medicinal plants are an important source of biologically active compounds. In the last four decades, 45% of newly approved antiparasitic drugs have been isolated or derived from natural products [9]. As such, plant secondary metabolites could provide new insights into discovering novel compounds against *T. gondii*. Our previous investigations demonstrated that a hexane fraction (Hf) obtained by solvent precipitation from *Pleopeltis crassinervata* (Pc) methanolic frond extract inhibited the in vitro growth of RH *T. gondii* strain tachyzoites with an IC_50_ of 16.9 µg/mL and was mostly positive for terpenoids in a phytochemical evaluation [10]. Therefore, in the present study, a bioguided analysis was performed to assess Pc subfraction activity against *T. gondii* as well as their cytotoxicity in order to obtain new leading compounds with anti-*Toxoplasma* activity, fewer adverse effects than conventional drugs, and activity against chronic toxoplasmosis. Chromatographic methods were used to identify compounds in subfraction number one.

## 2. Results

### 2.1. Fractionation of the Hexane Fraction by Thin-Layer Chromatography

Seven bands corresponding to each subfraction were obtained by thin-layer chromatography (TLC) (Figure 1). The eluent with the best resolution was a mixture of hexane/EtOAc at a ratio of 7:3. Subfractions were named according to elution order as hexane subfractions (Hsf) 1 to 7, with subfractions 1 and 4 having the highest extraction yields (33.3% and 14.28%, respectively) (Table 1). The average yield of Hsf1, with respect to the hexane fraction, was 44.28. In total, six batches of Hf were processed, and extraction yields were obtained (Table 2).

### 2.2. Subfraction Efficacy against T. gondii and Cytotoxicity

RH strain tachyzoites purified from infected BALB/c mice were used to evaluate the viability of treated parasites. The most active subfractions against *T. gondii* were Hsf1 and Hsf5, which affected tachyzoites at concentrations of 1 and 5 µg/mL, respectively. Parasites were completely killed when treated with Hsf1 at a concentration of 50 μg/mL; at the same concentration, Hsf4 and Hsf3 affected fewer than 50% and 77% of the treated tachyzoites, respectively. Tachyzoites were not affected by Hsf2, Hsf6, or Hsf7 (Figure 2). Parasites maintained in DMSO (0.02%) and untreated parasites showed 100% viability. The results are shown as the mean ± standard error of three independent studies (*p* < 0.05). The IC_50_ values for Hsf1, Hsf4, and Hsf5 were 23.69, 53.18, and 28.69 μg/mL, respectively (Table 3). Hsf1 showed the best activity against *T. gondii* tachyzoites and the highest extraction yield.

Based on the *T. gondii* viability results, we evaluated the cytotoxicity of Hsf1 and Hsf5 in Vero cells using resazurin dye and fluorimetry. The evaluated concentrations were the same as those used in the *T. gondii* viability assay and up to 800 μg/mL. Both fractions used in concentrations of up to 100 μg/mL did not affect cell cultures; however, at 200 μg/mL, cell cultures treated with Hsf1 and Hsf5 exhibited a viability of 87.93% and 87.72%, respectively, and with a concentration of 400 μg/mL, cell viability was 60.69% for Hsf1 and 69.51% for Hsf5. At the maximum concentration (800 μg/mL), viability values of 35.64% and 57.42% were observed after treatment with Hsf1 and Hsf5, respectively (Figure 3). The CC_50_ values for Hsf1 and Hsf5 were 398.7 and 394.4 μg/mL, respectively; the negative control DMSO (0.02%) did not affect cell culture viability. Additionally, SI for Hsf1 and HSf5 was calculated with respect to IC_50_ values. The results showed an SI of 16.89 for Hsf1 and 13.73 for Hsf5 (Figure 4). The SI for the reference drug pyrimethamine was 13.16, with an IC_50_ of 26 μg/mL and a CC_50_ of 342.23 μg/mL. The effect of *P. crassinervata* hexane subfractions on *T. gondii* tachyzoite viability as well as IC_50_ values are shown in Table 3.

### 2.3. Hsf1 Compound Identification by CG-MS

Four micrograms of Hsf1 were subjected to GC/MS analysis. The mass spectrum of the analyzed sample presented with 4 prominent peaks and 14 peaks under 9%; that is, these 4 peaks represent 52.684% of the total sample, and the rest of the compounds are present in a proportion of less than 9% each. (Figure 5). The major compounds identified in Hsf1 were hexadecanoid acid, methyl ester, olean-13(18)-ene, 2,2,4a,8a,9,12b,14a-octamethyl-1,2,3,4,4a,5,6,6a,6b,7,8,8a,9,12,12a,12b,13,14,14a,14b-eicosahydropicene, and 8-octadecenoid acid, methyl ester (Figure 6) in percentages of 18.05%, 16.19%, 12.99%, and 12.53%, respectively (Table 4). Some alkanes, such as dodecane, tetradecane, hexadecane, and octadecane, were also identified in Hsf1 at percentages lower than 5%.

## 3. Discussion

The search for new and efficient pharmacological alternatives for the treatment of toxoplasmosis is under intense and frequent evaluation. Different alternatives to sulfadiazine/pyrimethamine combinations have been reported, such as azithromycin, clarithromycin, spiramycin, atovaquone, dapsone, and trimethoprim-sulfamethoxazole; nevertheless, severe side effects, drug intolerance, poor compliance, malabsorption, and resistance have been associated with these therapies, and they are not effective against tissue cysts that are responsible for chronic infections. Bradyzoites contained in tissue cysts are capable of reactivating into the acute tachyzoite stage in immunocompromised hosts and cause severe complications such as encephalitis in AIDS patients [11,12].

Natural compounds seem to play an important role in discovering bioactive compounds with fewer side effects than actual synthetic drugs [13]. In the present study, the *P. crassinervata* hexane fraction was tested since previous results described its anti-*Toxoplasma* activity [10].

Even though plant extracts exhibited a great variety of secondary metabolites, when processing a fraction with low yields, we expected to find a low quantity of defined bands with TLC, which facilitated the obtainment of different subfractions [14]. In this study, seven defined bands were observed with TLC, and although each band would be expected to correspond to an isolated compound (as each band has a different retention time under the same chromatography conditions), defined bands do not always correspond to isolated compounds due to chemical similarities. Thus, one band could contain a mix of compounds with similar chemical features [15]. This was seen in Hsf1, which was composed of 18 compounds, including fatty acids and terpenes.

Hsf1 exhibited an IC_50_ of 23.6 µg/mL, a CC_50_ of 398.7 μg/mL, and an SI of 16.89; therefore, effective concentrations of Hsf1 against *T. gondii* are innocuous to host cells. In comparison, the reference drug pyrimethamine showed an IC_50_ of 26 μg/mL, a CC_50_ of 342.23 μg/mL, and an SI of 13.16. Apparently, Hsf1 is more effective, less cytotoxic, and safer than the reference drug.

Spiramycin and sulfadiazine are also used to treat toxoplasmosis, and their SI values are 0.72 and 1.15, respectively, in HeLa cells [16]. Our results showed that Hsf1 is fifteen times more specific and safer than these drugs. It has been reported that fatty acids in small amounts can disrupt the cell membrane and induce cell death [17]; however, Hsf1 did not affect Vero cell viability even at doses four times higher than the IC_50_. Fatty acids methyl esters (like some of those identified in Hsf1) with more hydrophilic head groups can reduce cytotoxicity [18]. Hexadecanoic acid, methyl ester (the compound found in the highest percentage in Hsf1), also known as methyl palmitate or hexadecanoate methyl ester, is a very hydrophobic molecule that is practically insoluble (in water) and relatively neutral. Its hydrophilicity possibly prevented a cytotoxic effect.

### 3.1. GC/MS Profiling

#### 3.1.1. Fatty Acids

Fatty acids are a broad group of amphipathic molecules with a carboxyl terminal group that, depending on the number of double bonds, are classified into saturated and unsaturated fatty acids. These molecules play a crucial role in the structure of cell membranes, energy production, signaling, and regulation of different biochemical routes [19]. In this study, the greatest anti-*Toxoplasma* activity was observed with Hsf1, which presented with the maximum extraction yield with respect to the hexane fraction. Seven fatty acids were present in Hsf1: one was polyunsaturated (arachidonic acid methyl ester) and the rest were saturated. Unsaturated fatty acids, commonly found at cell membranes, participate as second messengers that can alter intracellular and extracellular signaling pathways, affecting gene expression and physiologic and metabolic responses in different tissues [20]. Arachidonic acid has been isolated from *Laminaria digitata*, and its activity was evaluated against *Ascaris suum* and *Schistosoma mansoni*; moderate disintegration of parasite surface membranes and eventual parasite death were reported [21,22]. In this study, parasite cell membranes were affected by Hsf1, as shapeless treated tachyzoites were observed when dyed with Sytox green nucleic acid stain. Most likely, fatty acids from the active subfraction affected the lipid bilayer integrity or even the inner membrane complex of *T. gondii*. Both structures are vital for invasion and replication. Likewise, arachidonic acid metabolites seem to promote and modulate the Th2 immune response, which is important in resistance to parasites via direct action on eosinophils, basophils, and mast cells and indirectly by binding to specific receptors on innate lymphoid cells [23]. Docosahexaenoic acid, a polyunsaturated fatty acid, prevents *T. gondii* infection by inducing autophagy via AMP-activated protein kinase activation [24].

Long-chain saturated fatty acids contained in Hsf1 were hexadecanoic acid, methyl ester, 8-octadecenoic acid, methyl ester, octadecanoic acid, methyl ester, eicosanoid acid, methyl ester, docosanoic acid, methyl ester, and tetracosanoic acid, methyl ester. They all belong to fatty acid methyl esters (FAMEs), which are a type of fatty acid that are derived by the transesterification of fats with methanol and have been reported to exhibit antibacterial properties. Some authors have even described that FAMEs could be the next generation of antibiotic agents acting through membrane disruption and reactive oxygen species production [25].

Hexadecanoic acid, methyl ester was the major compound detected in Hsf1. Its activity against multidrug-resistant bacterial strains, such as *P. aeruginosa*, *B. subtilis*, and *K. pneumoniae*, has been reported; it seems that this fatty acid affects the integrity of bacterial cell membranes [26]. Likewise, hexadecanoic acid, methyl ester is the main constituent of diverse medicinal plants, such as *Gracilaria* spp., *Leucaena leucocephala*, and *Salix babylonica* [27,28]. It is suggested that this compound, in addition to affecting cell membranes, interferes with the production of cellular energy by inhibiting enzymatic activity. FAME can affect the fatty acid biosynthetic pathway by inhibiting FabI (enoyl-acyl carrier protein reductase) gene expression, and it has been proposed that peroxidation processes and inhibition of bacterial fatty acid synthesis are related to these molecules [26,29,30].

*T. gondii* depends on a type II fatty acid synthesis (FASII) different from the type I pathway observed in humans; a crucial enzyme involved in this pathway is Enoyl-acyl carrier protein reductase (ENR) [2]. The anti-*Toxoplasma* efficacy of some ENR inhibitors, such as triclosan analogs, has been reported [31]. Thiolactomycin affects the FAS II pathway, inhibiting the enzyme β-ketoacyl-acyl carrier protein synthase, which elongates fatty acids [32]. *T. gondii* acyl-CoA diacylglycerol acyltransferase (TgDGAT) is an integral membrane protein localized to the parasite cortical and perinuclear endoplasmic reticulum and is responsible for triacylglycerol synthesis. Thus, ablation of TgDGAT is lethal to the parasite. Some studies have reported that oleate, palmitoleate, and linoleate impair parasite replication by interfering with TgDGAT function [33]. Based on these experiences, fatty acids obtained from Hsf1 could disrupt parasite membranes by affecting oxidant–antioxidant homeostasis or inhibiting enzymes required in *T. gondii* fatty acid synthesis.

*T. gondii* evades the host immune response by different pathways, including the manipulation of host cell lipids and lipid-derived metabolites [33]; therefore, by controlling *T. gondii* lipidomics, parasite growth can be restricted, as previously reported by different authors [2].

#### 3.1.2. Terpenes

In recent results, we reported the presence of terpenes in the *Pleopletis crassinervata* hexane fraction from phytochemical screening [10]. These molecules are characterized by their isoprene unit number and have been extensively reported to exhibit antibiotic, anticancer, antioxidant, antiviral, antihyperglycemic, and antiparasitic properties [34,35,36].

Triterpenes, members of the broad group of terpenes, are characterized by the presence of six isoprene units and 30 carbon atoms, and some of these molecules exhibit biological activities such as anti-inflammatory (3β,6β,16β-trihydroxy-20(29)-eno), antinociceptive (oleanolic acid), antibiotic (Friedelin), and anticancer (20(S)-ginsenoside) effects [37].

A triterpenoid previously reported in natural products, olean-13(18)-ene, was detected in Hsf1. Some authors have reported the anti-inflammatory and antiproliferative activities of plant extracts containing this metabolite [38,39,40]. Commercially available triterpenes, such as betulin, betulinic acid, and betulone, have been shown to inhibit the growth of *T. gondii* RH tachyzoites with inhibition rates of 44%, 49%, and 99% at 10 mM, respectively. The n-heptane extract of *Alnus glutinosa* exhibited significant toxoplasmicidal activity without cytotoxicity and with an IC_50_ of 25.08 mg/mL and a selectivity index of 3.99; the major constituents of this extract were triterpenes [41]. Lupane-type triterpenoids extracted from the bark of black *Alnus glutinosa* exhibited in vitro anti-*Toxoplasma* (RH strain) activity. Betulone was the most active triterpene (IC_50_ of 2.7 µM, CC_50_ greater than 80 µM, and a selectivity index of over 29.6) [42]. Triterpenoids isolated from *Quercus crispula* Blume outer bark, 29-norlupane-3,20-dione, oleanolic acid acetate, and ursolic acid acetate, showed anti-*Toxoplasma* activity with an IC_50_ between 6.8 and 24.4 μM [43]. An *Olea europaea* pentacyclic triterpenoid, maslinic acid (2R,3_-dihydroxyolean-12-en-28-oic acid), was evaluated on *T. gondii* tachyzoites and showed an IC_50_ of 58.2 μM and a CC_50_ of 236 μM in Vero cells. It seems that this terpenoid disrupts parasite ultrastructure and motility due to an inhibition of protease activity [44]. Their capacity to affect lipid membranes has also been reported [45]. Tachyzoites affected by Hsf1 appeared shapeless, so this fraction probably disrupts parasite membranes.

Some pentacyclic triterpenes have been shown to inhibit the expression of COX-2 [46], which reduces *T. gondii* infection and upregulates the proinflammatory immune response in mice [47]. COX-2 is a crucial factor in *T. gondii* propagation in human trophoblasts; therefore, its inhibition can induce a proinflammatory response capable of controlling parasite proliferation [48]. Triterpenoids identified in Hsf1 are pentacyclic as well; therefore, they could affect the host Th1 immune response.

Another triterpenoid identified in Hsf1 was squalene, a precursor of cholesterol biosynthesis. *T. gondii* cannot synthetize cholesterol; hence, inhibitors of host squalene synthase can have interesting therapeutic potential, as they cut off cholesterol supplies, leading to *T. gondii* growth inhibition [49]. It has been reported that squalene enhances the immune response to various associated antigens, and it is therefore being investigated for vaccine delivery applications. Likewise, its antioxidant activity in cosmetic dermatology and anticancer properties in murine models have been studied [50].

#### 3.1.3. Alkanes

Alkanes such as dodecane, tetradecane, hexadecane and octadecane were also identified in Hsf1 at low percentages ranging from 1% to 4.4%. Alkanes are the simplest type of organic compounds that contain only carbon–carbon single bonds, and although there are no reports on their pharmacological activity, they have been widely detected in the essential oils of medicinal plants such as *Moringa peregrina, Euphorbia Heterophylla*, and *Olea europaea* [51,52,53]. Alkane aldehydes (E)-2-decenal, (E)-2-undecenal, and (E)-2-dodecenal present in *Coriandrum sativum* L. (Apiaceae) were reported to have in vitro antileishmanial activity [54]. The methanolic extract obtained from the hepatopancreas of *Halocynthia roretzi* (sea pineapple) showed antibacterial and antifungal activities against *Vibrio alginoliticus* and *Mortierella ranzuniana*. Bioassay-guided isolation resulted in a sulfated C9 alkane and three sulfated C10 alkenes (alkenes differ from alkanes by having one or more C=C double bonds) [55]. Alkane groups were found with spectroscopic analysis in the ethyl acetate extract of seaweed associated heterotrophic bacteria. The extract was active against vancomycin-resistant *Enterococcus faecalis* and methicillin-resistant *Staphylococcus aureus*, *Klebsiella pneumonia*, and *Pseudomonas aeruginosa* [56]. This group of compounds identified in Hsf1 should be studied carefully since alkanes are also important raw materials used in the chemical industry and are the principal constituent of gasoline and lubricating oils [51,52,53].

## 4. Materials and Methods

### 4.1. Plant Material

*Pleopeltis crassinervata* fronds were collected in the reproductive season (September-October) at Chignautla, Puebla, México (19°50′24.49″ N and 97°22′30.08″ O). A specimen was identified and deposited by Leticia Pacheco in the Metropolitan Herbarium (UAMIZ) at the Universidad Autonoma Metropolitana (UAM), CDMX with voucher number 84415. The fronds were dried at 37 °C for seven days and ground until use.

### 4.2. Extraction and Fractionation

*Pleopeltis crassinervata* fronds were collected and identified as previously reported by Anacleto-Santos et al., 2020, and the fronds were dried at 37 °C for seven days and ground until use [10].

### 4.3. Chromatographic Separation

The ground fronds were extracted in a solid–liquid–liquid system using methanol as a solvent for seven days and the methanolic extract was lyophilized. Subsequently it was dissolved in methanol and hexane in order to obtain methanolic, hexane, and precipitated fractions [57]. The hexane fraction was processed by column chromatography using a mixture of hexane/ethyl acetate at a 7:3 ratio as the eluent. The subfractions were collected and analyzed by thin-layer chromatography and named according to the order of elution.

### 4.4. Animals

Five-week-old BALB/c male mice were obtained from the vivarium at the School of Medicine, UNAM. Animal management was performed according to the Mexican Official Norm NOM-062-ZOO-1999 for the reproduction, care, and use of laboratory animals in accordance with international guidelines and approved by the Ethical and Research committee at the School of Medicine, UNAM (project 052/2017).

### 4.5. Cell Culture

To evaluate cytotoxic effects, African green monkey kidney epithelial cells (Vero CCL-81, ATCC) were maintained in Roswell Park Memorial Institute medium (RPMI-1640) supplemented with 10% inactivated fetal bovine serum (FBS) and antibiotics under a 5% CO_2_ atmosphere at 37 °C [58].

### 4.6. Parasites

RH strain tachyzoites, maintained by intraperitoneal (IP) passages in BALB/c mice, were harvested from the IP fluid of infected mice on the fifth day of infection and centrifuged twice (for ten minutes) at 260× *g*. Parasites were rinsed with phosphate-buffered saline (PBS) at pH 7.4, filtered through 5 μm membranes, and maintained in PBS. Purified tachyzoites were used within 2 h after isolation [59]. The RH strain was donated by CINVESTAV Zacatenco, Biochemistry Department, Intracellular Pathogens Laboratory, Mexico City.

### 4.7. Anti-Toxoplasma Activity Assay

Tachyzoite viability was evaluated in vitro by Sytox green^®^ dye exclusion. Purified tachyzoites (1 × 10^6^) were incubated in suspension with hexane subfractions that were first dissolved in 0.02% dimethyl sulfoxide (DMSO) and then diluted in serum-free MEM to final concentrations of 1–50 µg/mL for 1 h at room temperature. Then, Sytox green (1:500) was added and incubated for 15 min. The dye binds to parasite nucleic acids, emitting an exceptionally bright signal, and dead cell nucleic acids fluoresce bright green when excited with a wavelength of 450–490 nm [60]. Once bound to the genetic material, the fluorescence intensity of the dye is 500 times higher than when unbound in solution. Damaged parasites were counted using a fluorescence microscope. The experiment was performed in triplicate, meaning 300 tachyzoites were counted in independent events. Concentrations were determined according to the Hf IC_50_ values obtained in our previous studies [10]. Tachyzoites treated with pyrimethamine (10 µg/mL) and untreated tachyzoites maintained in PBS or 0.02% DMSO were used as control groups. The selectivity index (SI) was also calculated. SI is defined as the ratio of Hf cytotoxicity effects on uninfected cells divided by the anti-*Toxoplasma* activity; SI = CC_50_ Vero cells/IC_50_
*T. gondii* [61].

### 4.8. Gas Chromatography Analysis

Hsf1 was obtained by column chromatography. After evaporating the solvent under vacuum conditions, 4 µg were dissolved in DMSO and stored protected from light until use. The organic compounds present in Hsf1 were determined using gas chromatography (Agilent 6890 Plus) and mass spectrophotometry (Agilent 5973N; GC-MSD) systems. The GS-MS used in this study was carried out by external services provided by the Chemical Research Center, Autonomous University of state of Morelos (UAEM), Cuernavaca, Morelos Mexico, and was performed under operating conditions at a flow rate of 1 mL/min in split-less injection (1 µL) mode with an inlet temperature of 40 °C/10 min and an interface temperature of 250 °C with helium as the carrier gas. The compounds present in Hsf1 were identified by matching the GC-MS data with retention time (min), peak area, and mass spectral patterns from the mass spectral library NIST 1.7a [62].

### 4.9. Cytotoxic Assay

The Vero CCL-81 cell line was employed to determine Hsf1 cytotoxicity. Cells cultured in the logarithmic growth phase (1 × 10^3^ cells/well) were added to 96-well microplates in RPMI + 10% FBS and then exposed to 1–800 µg/mL of Hsf1 for 24 h at 37 °C and 5% CO_2_. After incubation, 20 µL of 1 µM resazurin solution was added to each well. The microplates were incubated at 37 °C in 5% CO_2_ for 3 h. DMSO and RPMI were used as negative controls. The viability percentage of Vero cells was determined by fluorometry, and absorbance readings were taken at 570 nm. The results were expressed as a percentage of cytotoxicity with respect to an untreated control; 0.02% DMSO, pyrimethamine, and a group without treatment were included as controls [63]. The experiment was carried out in triplicate independently.

### 4.10. Statistics

ANOVA and nonlinear regression analysis were used to analyze data (significance *p* < 0.05). Studies were performed in triplicate by independent evaluations.

## 5. Conclusions

We concluded that Pc Hsf1 is active against *T. gondii* RH tachyzoites and is not cytotoxic to Vero cells at concentrations that killed 100% of the parasites. Its efficacy could be related to the mechanisms of action of fatty acids and terpenes; nonetheless, as different molecules are found in this subfraction, a synergistic or antagonistic effect cannot be discarded. Hence, it is crucial to evaluate Hsf1 components as well as their mode of action, both in the parasite and host cells, to validate vital targets (preferably those that are not supplied by the host) leading to the identification of a lead molecule that will probably allow the design of analogous triterpenoids or FAMEs with anti-*Toxoplasma* activity. Likewise, the present study supports that this type of fern can be a good source of compounds with antimicrobial effects.

## Figures and Tables

**Figure 1 antibiotics-12-00889-f001:**
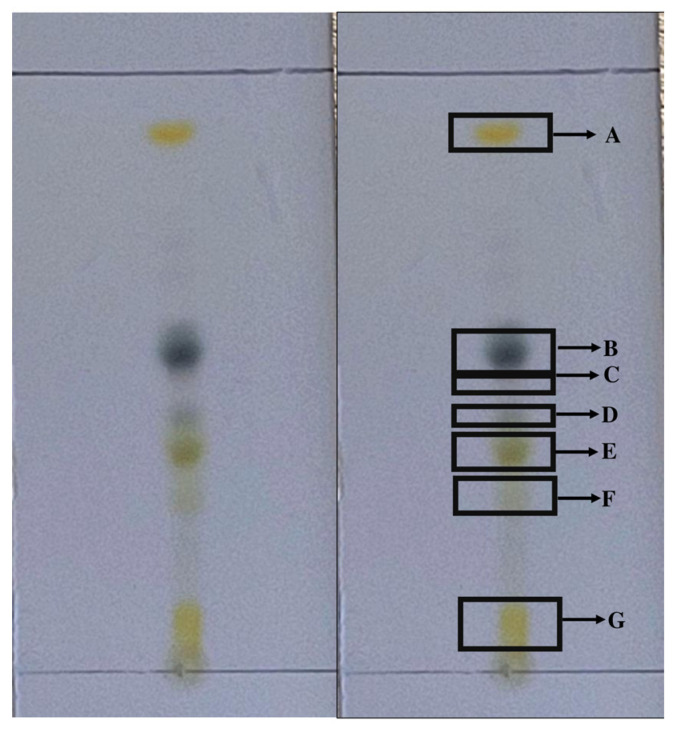
Subfractions obtained by thin-layer chromatography in a mobile phase of hexane/EtOAc 7:3 ratio. Each letter is a subfraction. A: Hsf1, B: Hsf2, C: Hsf3, D: Hsf4, E: Hsf5, F: Hsf6, and G: Hsf7. Hsf (hexane subfraction); EtOAc (Ethyl acetate).

**Figure 2 antibiotics-12-00889-f002:**
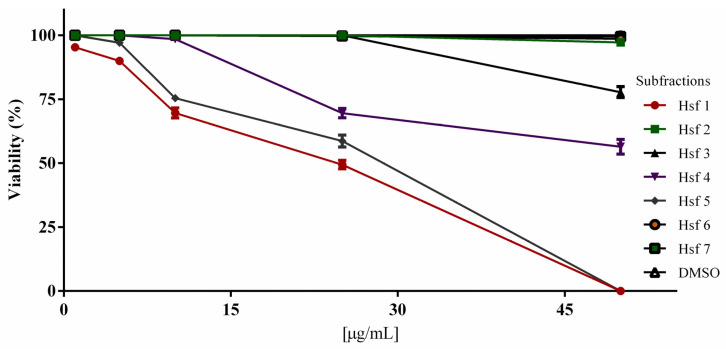
Tachyzoite viability percentage after one hour exposure to *Pleopeltis crassinervata* subfractions isolated by column chromatography. DMSO (dimethyl sulfoxide). Results are the mean ± standard error of three independent studies. *p* < 0.05.

**Figure 3 antibiotics-12-00889-f003:**
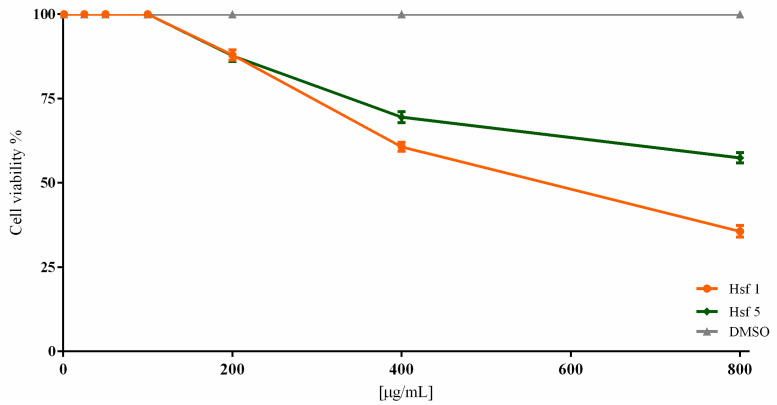
Viability percentages in Vero cell cultures exposed to Hsf1 and Hsf5 for 24 h in a concentration range of 25–800 µg/mL. A control with 0.02% DMSO was included, which showed 0% toxicity. Results are the mean ± standard error of three independent studies. *p* < 0.05. Hsf (hexane subfraction), DMSO (dimethyl sulfoxide).

**Figure 4 antibiotics-12-00889-f004:**
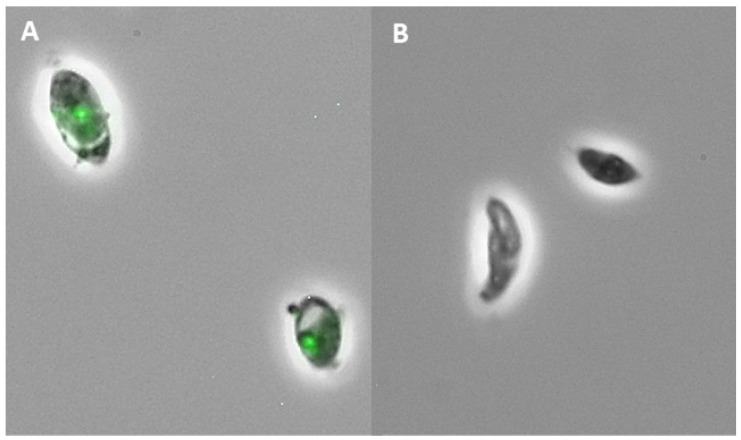
Photomicrographs of *Toxoplasma gondii* RH strain tachyzoites at 60 X. (**A**) Exposed to Hsf1 [50 µg/mL], observed on fluorescence microscopy; shapeless damaged parasites with green bright nucleus due to Sytox green^®^ dye. (**B**) Control without treatment in bright field on fluorescence microscopy; parasites appeared crescent shaped.

**Figure 5 antibiotics-12-00889-f005:**
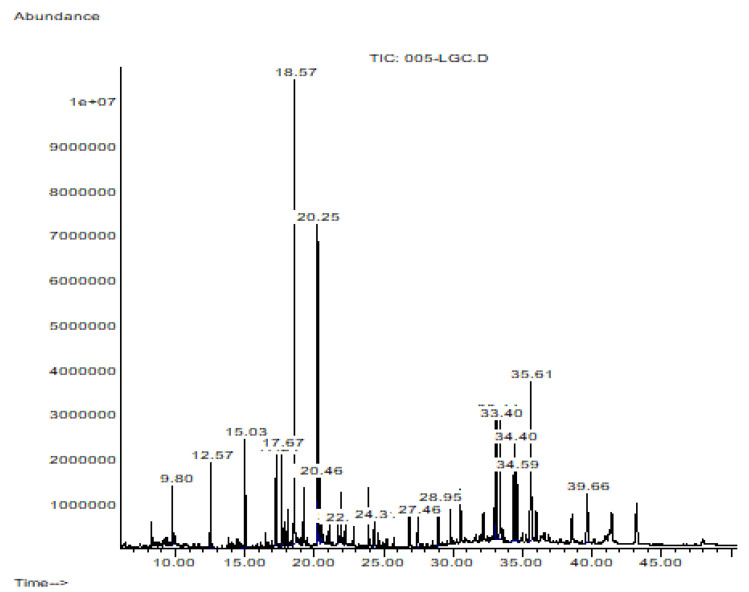
Hexane subfraction 1 chromatogram obtained by gas–mass chromatography (GS-MS) analysis. Each peak corresponds to the spectrum of each compound; 18 peaks were detected.

**Figure 6 antibiotics-12-00889-f006:**
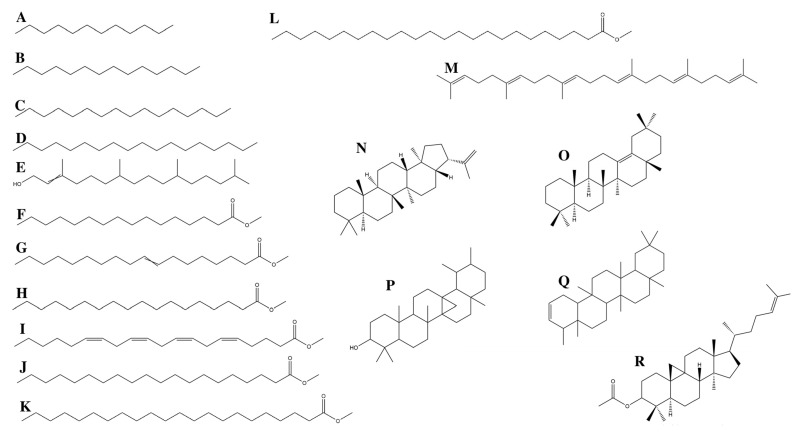
Chemical structures of hexane subfraction 1 (Hsf1) compounds; (**A**): dodecane, (**B**): tetradecane, (**C**): hexadecane, (**D**): octadecane, (**E**): 3,7,11,15-tetramethyl-2-hexadecen-1-ol, (**F**): hexadecanoic acid, methyl ester, (**G**): 8-octadecenoic acid, methyl ester, (**H**): octadecanoic acid, methyl ester, (**I**): 5,8,11,14-eicpsatetraenoic acid, methyl ester, (all-Z)-, (**J**): eicosanoid acid, methyl ester, (**K**): docosanoic acid, methyl ester, (**L**): tetracosanoic acid, methyl ester, (**M**): squalene, (**N**): A′-neogammacer-22(29)-ene, (**O**): olean-13(18)-ene, (**P**): 6a,14a-methanopicene, perhydro-1,2,4a,6b,9,9,12a-heptamethyl-10-hydroxy, (**Q**): 2,2,4a,8a,9,12b,14a-octamethyl-1,2,3,4,4a,5,6,6a,6b,7,8,8a,9,12,12a,12b,13,14,14a,14b-eicosahydropicene, (**R**): 9,19-cyclolanost-24-en-3-ol, acetate. Created with Chem draw 21.0.0 by J.A-S.

**Table 1 antibiotics-12-00889-t001:** Yields obtained from isolated subfractions (mg for each 500 mg of sample).

Subfraction	Hsf1	Hsf2	Hsf3	Hsf4	Hsf5	Hsf6	Residual
Yield mg/500 mg Hexane fraction	167.3	42.5	11.4	71.4	27.8	42.8	130
Yield %	33.3	8.5	2.28	14.28	5.56	8.84	26

Hsf (hexane subfraction).

**Table 2 antibiotics-12-00889-t002:** Subfraction yields from different batches.

Batch	1	2	3	4	5	6
Hexane fraction (g)	1. 5409	1.3013	1.5087	1.7367	1.4950	1.8501
Active subfraction (g)	0.8511	0.4132	0.7375	0.6338	0.5968	0.9885
Hsf1 yield (%)	55.23	31.75	48.88	36.49	39.91	53.42

Hsf (hexane subfraction).

**Table 3 antibiotics-12-00889-t003:** Efficacy of *Pleopeltis crassinervata* subfractions on *Toxoplasma gondii* tachyzoite viability.

Sample	1 µg/mL	5 µg/mL	10 µg/mL	25 µg/mL	50 µg/mL	µg/mL
	E (%)	E (%)	E (%)	E (%)	E (%)	IC_50_
Hsf1	95.35 ± 1.52	90 ± 2.18	69.67 ± 3.38	49.45 ± 2.83	0	**23.69**
Hsf2	100	100	100	100	96.12 ± 1.83	NC
Hsf3	100	100	100	100	77.8 ± 3.67	NC
Hsf4	100	100	98.6 ± 1.71	69.56 ± 3.16	56.45 ± 4.99	53.18
Hsf5	100	97.12 ± 1.16	75.45 ± 2.36	58.57 ± 4.04	0	28.69
Hsf6	100	100	100	99.7 ± 1.20	98.67 ± 0.38	NC
Hsf7	100	100	100	99.7 ± 0.38	99.55 ± 0.50	NC

The values obtained with the most active subfractions are shown in bold. Tachyzoites treated with DMSO [0.02%] and the untreated group showed 100% viability. Results are the mean ± standard error of three independent studies. *p* < 0.05. NC value was not possible to assess due to low activity. Hsf (hexane subfractions), E (efficacy), IC_50_ (mean inhibitory concentration).

**Table 4 antibiotics-12-00889-t004:** Chemical composition of hexane subfraction 1 (Hsf1) obtained by column chromatography and identified by gas–mass chromatography (GC/MS) analysis.

Elution Order	Compound	Retention Time	Formula	Area %	Synonyms
1	Dodecane	9.803	C_12_H_26_	1.795	Dihexyl
2	Tetradecane	12.568	C_14_H_30_	2.656	N- Tetradecane
3	Hexadecane	15.032	C_16_H_34_	4.492	etane
4	Octadecane	17.245	C_18_H_38_	2.395	n-Octadecane
5	3,7,11,15-Tetramethyl-2-hexadecen-1-ol	17.666	C_20_H_40_O	2.450	Phytol
6	Hexadecanoic acid, methyl ester	18.566	C_17_H_34_O_2_	18.054	Methyl Palmitate
7	8-Octadecenoic acid, methyl ester	20.247	C_19_H_36_O_2_	12.535	Methyl 8-octadecenoate
8	Octadecanoic acid, methyl ester	20.464	C_19_H_38_O_2_	1.910	Methyl Estereate
9	5,8,11,14-Eicpsatetraenoic acid, methyl ester, (all-Z)-	21.679	C_21_H_34_O_2_	0.606	Arachidonic acid methyl ester
10	Eicosanoid acid, methyl ester	22.224	C_21_H_42_O_2_	0.677	Methyl arachisate
11	Docosanoic acid, methyl ester	24.313	C_23_H_46_O_2_	1.335	Behenic acid, methyl ester
12	Tetracosanoic acid, methyl ester	27.459	C_25_H_50_O_2_	2.327	Methyl lignocerate
13	Squalene	28.951	C_30_H_50_	2.215	Spinacene
14	A-Neogammacer-22(29)-ene	33.109	C_30_H_50_	7.219	Diploptene
15	Olean-13(18)-ene	33.404	C_30_H_50_	16.194	-
16	6a,14a-methanopicene, perhydro-1,2,4a,6b,9,9,12a-heptamethyl-10-hydroxy	34.586	C_30_H_5_O	4.723	-
17	2,2,4a,8a,9,12b,14a-Octamethyl-1,2,3,4,4a,5,6,6a,6b,7,8,8a,9,12,12a,12b,13,14,14a,14b-eicosahydropicene	35.611	C_30_H_50_	12.996	-
18	9,19-Cyclolanost-24-en-3-ol, acetate	39.664	C_32_H_5_O_2_	5.422	Cycloartenol acetate

## Data Availability

Not applicable.

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
