# Peer review of "Evaluation of the Anti-Toxoplasma gondii Efficacy, Cytotoxicity, and GC/MS Profile of Pleopeltis crassinervata Active Subfractions"

_antibiotics, 2023, doi:10.3390/antibiotics12050889_

Round 1

Reviewer 1 Report

Comments for editors and authors

An interesting paper. In general article is well written. The data presented and the results obtained are certainly valuable and will add to the knowledge of scientific community. The analysed problems are clearly presented. Title adequately reflects the subject of the study. The manuscript is perfectly prepared. Summarizing, the data presented in reviewed manuscript are certainly worth publishing, but some corrections are recommended as below:

  1. Page 2, row 67 – for the first time should be “hexane fraction (Hf)” instead “Hf”
  2. Page 2, row 67 – analogically should be “Pleopaltis crassinervata (Pc) methanolic frond extract” instead “Pc methanolic frond extract.” Full name should be used for the first time in the Text, not only in Abstract.
  3. Each figure and/or each table should be clear separately, independently of the text and another tables or figures. Therefore titles of tables should be corrected. Full names of species should be presented; all abbreviations and/or symbols should be explained as footnotes. For example - Figure 2 – should be “Pleopaltis crassinervata” instead “P. crassinervata”; Figure 4 – should be “Toxoplasma gondii” instead “T. gondii;” Figures 1, 3, 5, 6, and Tables 3,  4 are not clear, all symbols should be explained.

Author Response

Comments Reviewer 1

  1. Page 2, row 67 – for the first time should be “hexane fraction (Hf)” instead “Hf”
  2. Page 2, row 67 – analogically should be “Pleopaltis crassinervata (Pc) methanolic frond extract” instead “Pc methanolic frond extract.” Full name should be used for the first time in the Text, not only in Abstract.
  3. Each figure and/or each table should be clear separately, independently of the text and another tables or figures. Therefore titles of tables should be corrected. Full names of species should be presented; all abbreviations and/or symbols should be explained as footnotes. For example - Figure 2 – should be “Pleopaltis crassinervata” instead “P. crassinervata”; Figure 4 – should be “Toxoplasma gondii” instead “T. gondii;” Figures 1, 3, 5, 6, and Tables 3,  4 are not clear, all symbols should be explained.

Response

The corrections have been made in the manuscript (highlighted in yellow).

Reviewer 2 Report

The study titled:"Evaluation of the anti-Toxoplasma gondii efficacy, cytotoxicity, and GC/MS profile of Pleopeltis crassinervata active subfractions" by Anacleto-Santos et al., examines the activity of different subfractions of Pleopeltis crassinervata (Pc), a fern used in traditional Mexican medicine for treating gastrointestinal ailments. The study indicates that the hexane fraction (Hf) obtained from Pc methanolic frond extract affects Toxoplasma gondii tachyzoite viability in vitro. The study contains some important findings however some major revisions are required prior to considering the paper for publication in Antibiotics.

Introduciton section:

The introduction lacks clarity and conciseness. The opening sentence stating the prevalence of Toxoplasma gondii infection in the world's population lacks context and does not explain why this information is important. The citation is also missing, indicating that the author has not done their due diligence in research. Additionally, the introduction fails to provide a clear research question or objective, leaving the reader unsure of the purpose of the study. Overall, this introduction is inadequate and requires significant revision.

Materials and Methods section:

Gaz Chromatography Analysis

It would be helpful to add the following information to enhance our understanding of the findings

  1. It would be helpful to know how the Hsf1 sample was prepared prior to gas chromatography and mass spectrophotometry analysis. Were any extractions or purification steps performed?

  2. Calibration standards: Were any calibration standards used to ensure the accuracy of the GC-MSD system? If so, what were they and how were they prepared?

  3. It would be useful to know how many replicates were analyzed and whether any statistical analyses were performed to evaluate the variability of the data.

Some grammatical errors need to be addressed in the revised version: 

  1. Line 161, "related to these therapies" should be changed to "associated with these therapies" for better clarity.
  2. Line 163, "reactivating" should be changed to "reactivate" to match the tense of the sentence.
  3. Line 186, "reduce the cell permeability barrier" should be changed to "disrupt the cell membrane" for better clarity.

Author Response

Comments Reviewer 2

Introduciton section:

The introduction lacks clarity and conciseness. The opening sentence stating the prevalence of Toxoplasma gondii infection in the world's population lacks context and does not explain why this information is important. The citation is also missing, indicating that the author has not done their due diligence in research. Additionally, the introduction fails to provide a clear research question or objective, leaving the reader unsure of the purpose of the study. Overall, this introduction is inadequate and requires significant revision.

Response

We have added more relevant information, citation and a clearer justification for conducting the study (highlighted in yellow in the manuscript).

Materials and Methods section:

Gaz Chromatography Analysis

It would be helpful to add the following information to enhance our understanding of the findings

  1. It would be helpful to know how the Hsf1 sample was prepared prior to gas chromatography and mass spectrophotometry analysis. Were any extractions or purification steps performed?

Response

4.3. Chromatographic separation

The ground fronds were extracted in a solid–liquid-liquid system using methanol as a solvent for seven days and the methanolic extract was lyophilized. Subsequently it was dissolved in methanol and hexane in order to obtain methanolic, hexane and precipitated 327 fractions [56]. The hexane fraction was processed by column chromatography using a mixture of hexane/ethyl acetate 7:3 ratio as eluent. The subfractions were collected and analyzed by thin layer chromatography and named according to the order of elution.

4.8. Gas Chromatography Analysis

Hsf1 was obtained by column chromatography, being the first subfraction to elute, after evaporating the solvent under vacuum conditions, 4 ug were dissolved in DMSO and stored protected from light until use. The organic compounds present in Hsf1 were determined using gas chromatography (Agilent 6890 Plus) and mass spectrophotometry (Agilent 5973N; GC–MSD) systems. The GS-MS used in this study was carried out by external services provided by the Chemical Research Center, Autonomous University of state of Morelos (UAEM), Cuernavaca, Morelos Mexico, and performed under operating conditions at a flow rate of 1 mL/min insplit-less injection (1 μL) mode with an inlet temperature of 40 °C/10 min and an interface temperature of 250 °C with helium as the carrier gas. The compounds present in Hsf1 were identified by matching the GC–MS data with retention time (min), peak area and mass spectral patterns from the mass spectral library NIST 1.7a [61].

All changes have been added to the manuscript (highlighted in yellow).

  1. Calibration standards: Were any calibration standards used to ensure the accuracy of the GC-MSD system? If so, what were they and how were they prepared?

Response

No calibration standards were used in the GC/MS analysis, the spectra obtained along with retention time and peak areas were only compared to the NIST 1.7a database mass spectral library.

  1. It would be useful to know how many replicates were analyzed and whether any statistical analyses were performed to evaluate the variability of the data.

Response

4.10. Statistics

ANOVA and nonlinear regression analysis were used to analyzed data (significance P < 0.05). Studies were performed in triplicate by independent evaluations.

All changes have been added to the manuscript (highlighted in yellow).

  1. Some grammatical errors need to be addressed in the revised version: 
    1. Line 161, "related to these therapies" should be changed to "associated with these therapies" for better clarity.
  • Line 163, "reactivating" should be changed to "reactivate" to match the tense of the sentence.
  • Line 186, "reduce the cell permeability barrier" should be changed to "disrupt the cell membrane" for better clarity.

All grammatical errors have been corrected in the manuscript (highlighted in yellow).

Reviewer 3 Report

REVIEW

Dear authors,

Please consider the following comments to improve the content of your manuscript before publication. 

The work provides information about the potential anti-Toxoplasma gondii effect of subfractions isolated from the Pleopeltis crassineervata plant, which is used in traditional Mexican medicine as a treatment for gastrointestinal ailments.

They meet the objective, although it is mandatory to demonstrate the anti-Toxoplasma gondii effect in an animal model, as well as to use a genotype II strain of T. gondii (tissue cyst-former), to demonstrate whether these subfractions of P. crassinervata on the latency phase of the parasite.

It is necessary to make the following corrections in the indicated lines:

Lines 26, 33, 35, 54, 58, 167, 168, 200, 239, 270, 274, 396: write in cursive “anti-Toxoplasma”.

Line 39: write the abbreviation in “Toxoplasma gondii (T. gondii)”.

Lines 68, 349: write in cursive “in vitro”.

Lines 167, 206, 267, 272, 273, 283, 321, 389: write in cursive “P. crassinervata”, “Laminaria digitata”, “Alnus glutinosa”, “Quercus crispula”, “T. gondii”, “Pleopeltis crassinervata”.

Line 323: delete point in “use.[9].”.

Line 339: write in subscript “CO2”.

Please amend the requested comments and submit the revision file.

Author Response

Comments Reviewer 3

It is necessary to make the following corrections in the indicated lines:

 Lines 26, 33, 35, 54, 58, 167, 168, 200, 239, 270, 274, 396: write in cursive “anti-Toxoplasma”.

Line 39: write the abbreviation in “Toxoplasma gondii (T. gondii)”.

Lines 68, 349: write in cursive “in vitro”.

Lines 167, 206, 267, 272, 273, 283, 321, 389: write in cursive “P. crassinervata”, “Laminaria digitata”, “Alnus glutinosa”, “Quercus crispula”, “T. gondii”, “Pleopeltis crassinervata”.

Line 323: delete point in “use.[9].”.Line 339: write in subscript “CO2”.

Response

The corrections have been made in the manuscript (highlighted in yellow).

Round 2

Reviewer 2 Report

Great job to the authors for effectively addressing all of my inquiries, which now makes the manuscript suitable for publication in its present state.